# Robust and Efficient Transfer Learning with Hidden Parameter Markov Decision Processes

**Taylor Killian**[*]
taylorkillian@g.harvard.edu
Harvard University

**Samuel Daulton**[*]
sdaulton@g.harvard.edu
Harvard University, Facebook[†]

**George Konidaris**
gdk@cs.brown.edu
Brown University

**Finale Doshi-Velez**
finale@seas.harvard.edu
Harvard University

## Abstract

We introduce a new formulation of the Hidden Parameter Markov Decision Process (HiP-MDP), a framework for modeling families of related tasks using low-dimensional latent embeddings. Our new framework correctly models the joint uncertainty in the latent parameters and the state space. We also replace the original Gaussian Process-based model with a Bayesian Neural Network, enabling more scalable inference. Thus, we expand the scope of the HiP-MDP to applications with higher dimensions and more complex dynamics.

## 1 Introduction

The world is filled with families of tasks with similar, but not identical, dynamics. For example, consider the task of training a robot to swing a bat with unknown length $l$ and mass $m$. The task is a member of a family of bat-swinging tasks. If a robot has already learned to swing several bats with various lengths and masses $\{(l_i, m_i)\}_{i=1}^N$, then the robot should learn to swing a new bat with length $l'$ and mass $m'$ more efficiently than learning from scratch. That is, it is grossly inefficient to develop a control policy from scratch each time a unique task is encountered.

The Hidden Parameter Markov Decision Process (HiP-MDP) [14] was developed to address this type of transfer learning, where optimal policies are adapted to subtle variations within tasks in an efficient and robust manner. Specifically, the HiP-MDP paradigm introduced a low-dimensional latent task parameterization $w_b$ that, combined with a state and action, completely describes the system's dynamics $T(s'|s, a, w_b)$. However, the original formulation did not account for nonlinear interactions between the latent parameterization and the state space when approximating these dynamics, which required all states to be visited during training. In addition, the original framework scaled poorly because it used Gaussian Processes (GPs) as basis functions for approximating the task's dynamics.

We present a new HiP-MDP formulation that models interactions between the latent parameters $w_b$ and the state $s$ when transitioning to state $s'$ after taking action $a$. We do so by including the latent parameters $w_b$, the state $s$, and the action $a$ as *input* to a Bayesian Neural Network (BNN). The BNN both learns the common transition dynamics for a family of tasks and models how the unique variations of a particular instance impact the instance's overall dynamics. Embedding the latent parameters in this way allows for more accurate uncertainty estimation and more robust transfer when learning a control policy for a new and possibly unique task instance. Our formulation also inherits several desirable properties of BNNs: it can model multimodal and heteroskedastic transition

---

[*]Both contributed equally as primary authors
[†]Current affiliation, joined afterward

functions, inference scales to data large in both dimension and number of samples, and all output dimensions are jointly modeled, which reduces computation and increases predictive accuracy [11]. Herein, a BNN can capture complex dynamical systems with highly non-linear interactions between state dimensions. Furthermore, model uncertainty is easily quantified through the BNN's output variance. Thus, we can scale to larger domains than previously possible.

We use the improved HiP-MDP formulation to develop control policies for acting in a simple two-dimensional navigation domain, playing acrobot [42], and designing treatment plans for simulated patients with HIV [15]. The HiP-MDP rapidly determines the dynamics of new instances, enabling us to quickly find near-optimal instance-specific control policies.

## 2   Background

**Model-based reinforcement learning**   We consider reinforcement learning (RL) problems in which an agent acts in a continuous state space $S \subseteq \mathbb{R}^D$ and a discrete action space $A$. We assume that the environment has some true transition dynamics $T(s'|s, a)$, unknown to the agent, and we are given a reward function $R(s, a) : S \times A \to \mathbb{R}$ that provides the utility of taking action $a$ from state $s$. In the model-based reinforcement learning setting, our goal is to learn an approximate transition function $\hat{T}(s'|s, a)$ based on observed transitions $(s, a, s')$ and then use $\hat{T}(s'|s, a)$ to learn a policy $a = \pi(s)$ that maximizes long-term expected rewards $E[\sum_t \gamma^t r_t]$, where $\gamma \in (0, 1]$ governs the relative importance of immediate and future rewards.

**HiP-MDPs**   A HiP-MDP [14] describes a *family* of Markov Decision Processes (MDPs) and is defined by the tuple $\{S, A, W, T, R, \gamma, P_W\}$, where $S$ is the set of states $s$, $A$ is the set of actions $a$, and $R$ is the reward function. The transition dynamics $T(s'|s, a, w_b)$ for each task instance $b$ depend on the value of the hidden parameters $w_b \in W$; for each instance, the parameters $w_b$ are drawn from prior $P_W$. The HiP-MDP framework assumes that a finite-dimensional array of hidden parameters $w_b$ can fully specify variations among the true task dynamics. It also assumes the system dynamics are invariant during a task and the agent is signaled when one task ends and another begins.

**Bayesian Neural Networks**   A Bayesian Neural Network (BNN) is a neural network, $f(\cdot, \cdot; \mathcal{W})$, in which the parameters $\mathcal{W}$ are random variables with some prior $P(\mathcal{W})$ [27]. We place independent Gaussian priors on each parameter $P(\mathcal{W}) = \prod_{w \in \mathcal{W}} \mathcal{N}(w; \mu, \sigma^2)$. Exact Bayesian inference for the posterior over parameters $P(\mathcal{W}|\{(s', s, a)\})$ is intractable, but several recent techniques have been developed to scale inference in BNNs [4, 17, 22, 33]. As probabilistic models, BNNs reduce the tendency of neural networks to overfit in the presence of low amounts of data—just as GPs do. In general, training a BNN is more computationally efficient than a GP [22], while still providing coherent uncertainty measurements. Specifically, predictive distributions can be calculated by taking averages over samples of $\mathcal{W}$ from an approximated posterior distribution over the parameters. As such, BNNs are being adopted in the estimation of stochastic dynamical systems [11, 18].

## 3   A HiP-MDP with Joint-Uncertainty

The original HiP-MDP transition function models variation across task instances as:[3]

$$
\begin{aligned}
s'_d &\approx \sum_{k=1}^{K} w_{bk} \hat{T}_{kad}^{(GP)}(s) + \epsilon \\
w_{bk} &\sim \mathcal{N}(\mu_{w_k}, \sigma_w^2) \\
\epsilon &\sim \mathcal{N}(0, \sigma_{ad}^2),
\end{aligned}
\tag{1}
$$

where $s_d$ is the $d^{th}$ dimension of $s$. Each basis transition function $\hat{T}_{kad}$ (indexed by the $k^{th}$ latent parameter, the action $a$, and the dimension $d$) is a GP using only $s$ as input, linearly combined with instance-specific weights $w_{bk}$. Inference involves learning the parameters for the GP basis functions and the weights for each instance. GPs can robustly approximate stochastic state transitions in

continuous dynamical systems in model-based reinforcement learning [9, 35, 36]. GPs have also been widely used in transfer learning outside of RL (e.g. [5]).

While this formulation is expressive, it has limitations. The primary limitation is that the uncertainty in the latent parameters $w_{kb}$ is modeled independently of the agent's state uncertainty. Hence, the model does not account for interactions between the latent parameterization $w_b$ and the state $s$. As a result, Doshi-Velez and Konidaris [14] required that each task instance $b$ performed the *same set* of state-action combinations $(s, a)$ during training. While such training may sometimes be possible—e.g. robots that can be driven to identical positions—it is onerous at best and *impossible* for other systems such as human patients. The secondary limitation is that each output dimension $s_d$ is modeled separately as a collection of GP basis functions $\{\hat{T}_{kad}\}_{k=1}^K$. The basis functions for output dimension $s_d$ are independent of the basis functions for output dimension $s_{d'}$, for $d \neq d'$. Hence, the model does not account for correlation between output dimensions. Modeling such correlations typically requires knowledge of how dimensions interact in the approximated dynamical system [2, 19]. We choose not to constrain the HiP-MDP with such a priori knowledge since the aim is to provide basis functions that can ascertain these relationships through observed transitions.

To overcome these limitations, we include the instance-specific weights $w_b$ as input to the transition function and model all dimensions of the output jointly:

$$s' \approx \hat{T}^{(BNN)}(s, a, w_b) + \epsilon$$
$$w_b \sim \mathcal{N}\left(\mu_w, \Sigma_b\right)$$
$$\epsilon \sim \mathcal{N}\left(0, \sigma_n^2\right). \tag{2}$$

This critical modeling change eliminates *all* of the above limitations: we can learn *directly* from data as observed—which is abundant in many industrial and health domains—and no longer require highly constrained training procedure. We can also capture the correlations in the outputs of these domains, which occur in many natural processes.

Finally, the computational demands of using GPs as the transition function limited the application of the original HiP-MDP formulation to relatively small domains. In the following, we use a BNN rather than a GP to model this transition function. The computational requirements needed to learn a GP-based transition function makes a direct comparison to our new BNN-based formulation infeasible within our experiments (Section 5). We demonstrate, in Appendix A, that the BNN-based transition model *far* exceeds the GP-based transition model in both computational and predictive performance. In addition, BNNs naturally produce multi-dimensional outputs $s'$ without requiring prior knowledge of the relationships between dimensions. This allows us to directly model output correlations between the $D$ state dimensions, leading to a more unified and coherent transition model. Inference in a larger input space $s, a, w_b$ with a large number of samples is tractable using efficient approaches that let us—given a distribution $P(\mathcal{W})$ and input-output tuples $(s, a, s')$—estimate a distribution over the latent embedding $P(w_b)$. This enables more robust, scalable transfer.

**Demonstration**   We present a toy domain (Figure 1) where an agent is tasked with navigating to a goal region. The state space is continuous ($s \in (-2, 2)^2$), and action space is discrete ($a \in \{N, E, S, W\}$). Task instances vary the following the domain aspects: the location of a wall that blocks access to the goal region (either to the left of or below the goal region), the orientation of the cardinal directions (i.e. whether taking action North moves the agent up or down), and the direction of a nonlinear wind effect that increases as the agent moves away from the start region. Ignoring the wall and grid boundaries, the transition dynamics are:

$$\Delta x = (-1)^{\theta_b} c\big(a_x - (1 - \theta_b)\beta\sqrt{(x + 1.5)^2 + (y + 1.5)^2}\big)$$
$$\Delta y = (-1)^{\theta_b} c\big(a_y - \theta_b\beta\sqrt{(x + 1.5)^2 + (y + 1.5)^2}\big)$$
$$a_x = \begin{cases} 1 & a \in \{E, W\} \\ 0 & \text{otherwise} \end{cases}$$
$$a_y = \begin{cases} 1 & a \in \{N, S\} \\ 0 & \text{otherwise,} \end{cases}$$

where $c$ is the step-size (without wind), $\theta_b \in \{0, 1\}$ indicates which of the two classes the instance belongs to and $\beta \in (0, 1)$ controls the influence of the wind and is fixed for all instances. The agent

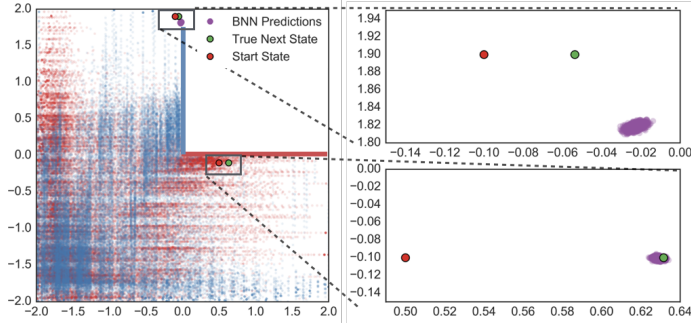

Figure 1: A demonstration of the HiPMDP modeling the joint uncertainty between the latent parameters $w_b$ and the state space. On the left, blue and red dots show the exploration during the red ($\theta_b = 0$) and blue ($\theta_b = 1$) instances. The latent parameters learned from the red instance are used predict transitions for taking action $E$ from an area of the state space either unexplored (top right) or explored (bottom right) during the red instance. The prediction variance provides an estimate of the joint uncertainty between the latent parameters $w_b$ and the state.

is penalized for trying to cross a wall, and each step incurs a small cost until the agent reaches the goal region, encouraging the agent to discover the goal region with the shortest route possible. An episode terminates once the agent enters the goal region or after 100 time steps.

A linear function of the state $s$ and latent parameters $w_b$ would struggle to model both classes of instances ($\theta_b = 0$ and $\theta_b = 1$) in this domain because the state transition resulting from taking an action $a$ is a nonlinear function with interactions between the state and hidden parameter $\theta_b$.

By contrast, our new HiP-MDP model allows nonlinear interactions between state and the latent parameters $w_b$, as well as jointly models their uncertainty. In Figure 1, this produces measurable differences in transition uncertainty in regions where there are few related observed transitions, even if there are many observations from unrelated instances. Here, the HiP-MDP is trained on two instances from distinct classes (shown in blue ($\theta_b = 1$) and red ($\theta_b = 0$) on the left). We display the uncertainty of the transition function, $\hat{T}$, using the latent parameters $w_{\text{red}}$ inferred for a red instance in two regions of the domain: 1) an area explored during red instances and 2) an area not explored under red instances, but explored with blue instances. The transition uncertainty $\hat{T}$ is three times larger in the region where red instances have not been—even if many blue instances have been there—than in regions where red instances have commonly explored, demonstrating that the latent parameters can have different effects on the transition uncertainty in different states.

## 4 Inference

Algorithm 1 summarizes the inference procedure for learning a policy for a new task instance $b$, facilitated by a pre-trained BNN for that task, and is similar in structure to prior work [9, 18]. The procedure involves several parts. Specifically, at the start of a new instance $b$, we have a global replay buffer $\mathcal{D}$ of all observed transitions $(s, a, r, s')$ and a posterior over the weights $\mathcal{W}$ for our BNN transition function $\hat{T}$ learned with data from $\mathcal{D}$. The first objective is to quickly determine the latent embedding, $w_b$, of the current instance's specific dynamical variation as transitions $(s, a, s')$ are observed from the current instance. Transitions from instance $b$ are stored in both the global replay buffer $\mathcal{D}$ and an instance-specific replay buffer $\mathcal{D}_b$. The second objective is to develop an optimal control policy using the transition model $\hat{T}$ and learned latent parameters $w_b$. The transition model $\hat{T}$ and latent embedding $w_b$ are separately updated via mini-batch stochastic gradient descent (SGD) using Adam [26]. Using $\hat{T}$ for planning increases our sample efficiency as we reduce interactions with the environment. We describe each of these parts in more detail below.

### 4.1 Updating embedding $w_b$ and BNN parameters $\mathcal{W}$

For each new instance, a new latent weighting $w_b$ is sampled from the prior $P_W$ (Alg. 1, step 2), in preparation of estimating unobserved dynamics introduced by $\theta_b$. Next, we observe transitions $(s, a, r, s')$ from the task instance for an initial exploratory episode (Alg. 1, steps 7-10). Given that

---
**Algorithm 1** Learning a control policy w/ the HiP-MDP
---

**Input:** Global replay buffer $\mathcal{D}$, BNN transition function $\hat{T}$, initial state $s_b^0$

1: **procedure** LEARNPOLICY( $\mathcal{D}, \hat{T}, s_b^0$)
2:     Draw new $w_b \sim P_W$
3:     Randomly init. policy $\hat{\pi}_b$ $\theta, \theta^-$
4:     Init. instance replay buffer $\mathcal{D}_b$
5:     Init. fictional replay buffer $\mathcal{D}_b^f$
6:     **for** $i = 0$ to $N_e$ episodes **do**
7:         **repeat**
8:             Take action $a \leftarrow \hat{\pi}_b(s)$
9:             Store $\mathcal{D}, \mathcal{D}_b \leftarrow (s, a, r, s', w_b)$
10:         **until** episode is complete
11:         **if** $i = 0$ OR $\hat{T}$ is innaccurate **then**
12:             $\mathcal{D}_b, \mathcal{W}, w_b \leftarrow$ TUNEMODEL($\mathcal{D}_b, \mathcal{W}, w_b$)
13:             **for** $j = 0$ to $N_f - 1$ episodes **do**
14:                 $\mathcal{D}_b^f, \hat{\pi}_b \leftarrow$ SIMEP($\mathcal{D}_b^f, \hat{T}, w_b, \hat{\pi}_b, s_b^0$)
15:         $\mathcal{D}_b^f, \hat{\pi}_b \leftarrow$ SIMEP($\mathcal{D}_b^f, \hat{T}, w_b, \hat{\pi}_b, s_b^0$)

1: **function** SIMEP($\mathcal{D}_b^f, \hat{T}, w_b, \hat{\pi}_b, s_b^0$)
2:     **for** $t = 0$ to $N_t$ time steps **do**
3:         Take action $a \leftarrow \hat{\pi}_b(s)$
4:         Approx. $\hat{s}' \leftarrow \hat{T}(s, a, w_b)$
5:         Calc. reward $\hat{r} \leftarrow R(s, a, \hat{s}')$
6:         Store $\mathcal{D}_b^f \leftarrow (s, a, \hat{r}, \hat{s}')$
7:         **if** $\mod (t, N_\pi) = 0$ **then**
8:             Update $\hat{\pi}_b$ via $\theta$ from $\mathcal{D}_b^f$
9:             $\theta^- \leftarrow \tau\theta + (1 - \tau)\theta^-$
10:     **return** $\mathcal{D}_b^f, \hat{\pi}_b$

1: **function** TUNEMODEL($\mathcal{D}_b, \mathcal{W}, w_b$)
2:     **for** $k = 0$ to $N_u$ updates **do**
3:         Update $w_b$ from $\mathcal{D}_b$
4:         Update $\mathcal{W}$ from $\mathcal{D}_b$
5:     **return** $\mathcal{D}_b, \mathcal{W}, w_b$

---

data, we optimize the latent parameters $w_b$ to minimize the $\alpha$-divergence of the posterior predictions of $\hat{T}(s, a, w_b | \mathcal{W})$ and the true state transitions $s'$ (step 3 in TuneModel) [22]. Here, the minimization occurs by adjusting the latent embedding $w_b$ while holding the BNN parameters $\mathcal{W}$ fixed. After an initial update of the $w_b$ for a newly encountered instance, the parameters $\mathcal{W}$ of the BNN transition function $\hat{T}$ are optimized (step 4 in TuneModel). As the BNN is trained on multiple instances of a task, we found that the only additional data needed to refine the BNN and latent $w_b$ for some new instance can be provided by an initial exploratory episode. Otherwise, additional data from subsequent episodes can be used to further improve the BNN and latent estimates (Alg. 1, steps 11-14).

The mini-batches used for optimizing the latent $w_b$ and BNN network parameters $\mathcal{W}$ are sampled from $\mathcal{D}_b$ with squared error prioritization [31]. We found that switching between small updates to the latent parameters and small updates to the BNN parameters led to the best transfer performance. If either the BNN network or latent parameters are updated too aggressively (having a large learning rate or excessive number of training epochs), the BNN disregards the latent parameters or state inputs respectively. After completing an instance, the BNN parameters and the latent parameters are updated using samples from global replay buffer $\mathcal{D}$. Specific modeling details such as number of epochs, learning rates, etc. are described in Appendix C.

## 4.2 Updating policy $\hat{\pi}_b$

We construct an $\varepsilon$-greedy policy to select actions based on an approximate action-value function $\hat{Q}(s, a)$. We model the action value function $\hat{Q}(s, a)$ with a Double Deep Q Network (DDQN) [21, 29]. The DDQN involves training two networks (parametrized by $\theta$ and $\theta^-$ respectively), a primary Q-network, which informs the policy, and a target Q-network, which is a slowly annealed copy of the primary network (step 9 of SimEp) providing greater stability when updating the policy $\hat{\pi}_b$ .

With the updated transition function, $\hat{T}$, we approximate the environment when developing a control policy (SimEp). We simulate batches of entire episodes of length $N_t$ using the approximate dynamical model $\hat{T}$, storing each transition in a fictional experience replay buffer $\mathcal{D}_b^f$ (steps 2-6 in SimEp). The primary network parameters $\theta$ are updated via SGD every $N_\pi$ time steps (step 8 in SimEp) to minimize the temporal-difference error between the primary network's and the target network's Q-values. The mini-batches used in the update are sampled from the fictional experience replay buffer $\mathcal{D}_b^f$, using TD-error-based prioritization [38].

# 5 Experiments and Results

Now, we demonstrate the performance of the HiP-MDP with embedded latent parameters in transferring learning across various instances of the same task. We revisit the 2D demonstration problem from Section 3, as well as describe results on both the acrobot [42] and a more complex healthcare domain: prescribing effective HIV treatments [15] to patients with varying physiologies.[4]

For each of these domains, we compare our formulation of the HiP-MDP with embedded latent parameters (equation 2) with four baselines (one model-free and three model-based) to demonstrate the efficiency of learning a policy for a new instance $b$ using the HiP-MDP. These comparisons are made across the first handful of episodes encountered in a new task instance to highlight the advantage provided by transferring information through the HiP-MDP. The 'linear' baseline uses a BNN to learn a set of basis functions that are linearly combined with the parameters $w_b$ (used to approximate the approach of Doshi-Velez and Konidaris [14], equation 1), which does not allow interactions between states and weights. The 'model-based from scratch' baseline considers each task instance $b$ as unique; requiring the BNN transition function to be trained only on observations made from the current task instance. The 'average' model baseline is constructed under the assumption that a single transition function can be used for every instance of the task; $\hat{T}$ is trained from observations of all task instances together. For all model-based approaches, we replicated the HiP-MDP procedure as closely as possible. The BNN was trained on observations from a single episode before being used to generate a large batch of approximate transition data, from which a policy is learned. Finally, the model-free baseline learns a DDQN-policy directly from observations of the current instance.

For more information on the experimental specifications and long-run policy learning see Appendix C and D, respectively.

## 5.1 Revisiting the 2D demonstration

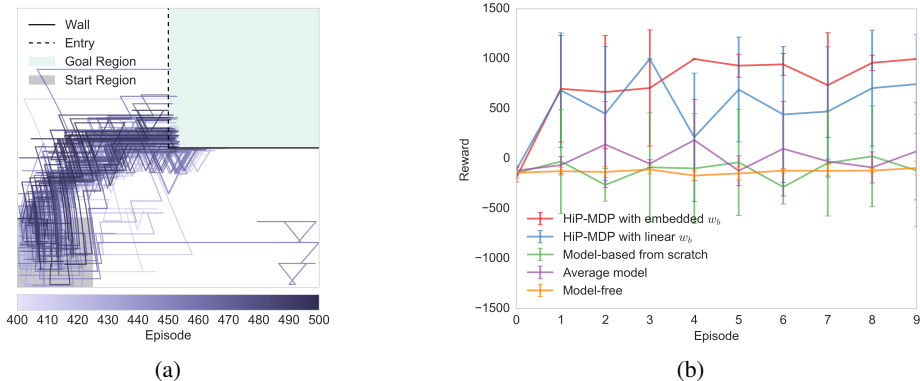

(a)  (b)

Figure 2: (a) a demonstration of a model-free control policy, (b) a comparison of learning a policy at the outset of a new task instance $b$ using the HiP-MDP versus four benchmarks. The HiP-MDP with embedded $w_b$ outperforms all four benchmarks.

The HiP-MDP and the average model were supplied a transition model $\hat{T}$ trained on two previous instances, one from each class, before being updated according to the procedure outlined in Sec. 4 for a newly encountered instance. After the first exploratory episode, the HiP-MDP has sufficiently determined the latent embedding, evidenced in Figure 2b where the developed policy clearly outperforms all four benchmarks. This implies that the transition model $\hat{T}$ adequately provides the accuracy needed to develop an optimal policy, aided by the learned latent parametrization.

The HiP-MDP with linear $w_b$ also quickly adapts to the new instance and learns a good policy. However, the HiP-MDP with linear $w_b$ is unable to model the nonlinear interaction between the latent parameters and the state. Therefore the model is less accurate and learns a less consistent policy than the HiP-MDP with embedded $w_b$. (See Figure 2a in Appendix A.2)

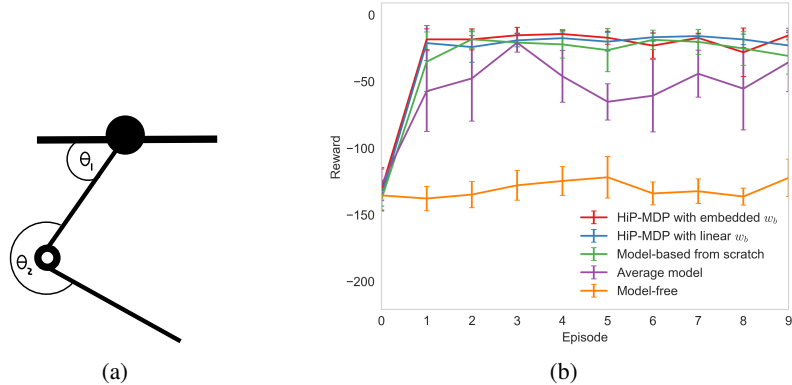

(a)                                    (b)

Figure 3: (a) the acrobot domain, (b) a comparison of learning a policy for a new task instance $b$ using the HiP-MDP versus four benchmarks.

With single episode of data, the model trained from scratch on the current instance is not accurate enough to learn a good policy. Training a BNN from scratch requires more observations of the true dynamics than are necessary for the HiP-MDP to learn the latent parameterization and achieve a high level of accuracy. The model-free approach eventually learns an optimal policy, but requires significantly more observations to do so, as represented in Figure 2a. The model-free approach has no improvement in the first 10 episodes. The poor performance of the average model approach indicates that a single model cannot adequately represent the dynamics of the different task instances. Hence, learning a latent representation of the dynamics specific to each instance is crucial.

## 5.2 Acrobot

First introduced by Sutton and Barto [42], acrobot is a canonical RL and control problem. The most common objective of this domain is for the agent to swing up a two-link pendulum by applying a positive, neutral, or negative torque on the joint between the two links (see Figure 3a). These actions must be performed in sequence such that the tip of the bottom link reaches a predetermined height above the top of the pendulum. The state space consists of the angles $\theta_1$, $\theta_2$ and angular velocities $\dot{\theta}_1$, $\dot{\theta}_2$, with hidden parameters corresponding to the masses ($m_1$, $m_2$) and lengths ($l_1$, $l_2$), of the two links.[5] See Appendix B.2 for details on how these hidden parameters were varied to create different task instances. A policy learned on one setting of the acrobot will generally perform poorly on other settings of the system, as noted in [3]. Thus, subtle changes in the physical parameters require separate policies to adequately control the varied dynamical behavior introduced. This provides a perfect opportunity to apply the HiP-MDP to transfer between separate acrobot instances when learning a control policy $\hat{\pi}_b$ for the current instance.

Figure 3b shows that the HiP-MDP learns an optimal policy after a single episode, whereas all other model-based benchmarks required an additional episode of training. As in the toy example, the model-free approach eventually learns an optimal policy, but requires more time.

## 5.3 HIV treatment

Determining effective treatment protocols for patients with HIV was introduced as an RL problem by mathematically representing a patient's physiological response to separate classes of treatments [1, 15]. In this model, the state of a patient's health is recorded via 6 separate markers measured with a blood test.[6] Patients are given one of four treatments on a regular schedule. Either they are given treatment from one of two classes of drugs, a mixture of the two treatments, or provided no treatment (effectively a rest period). There are 22 hidden parameters in this system that control a patient's specific physiology and dictate rates of virulence, cell birth, infection, and death. (See Appendix B.3

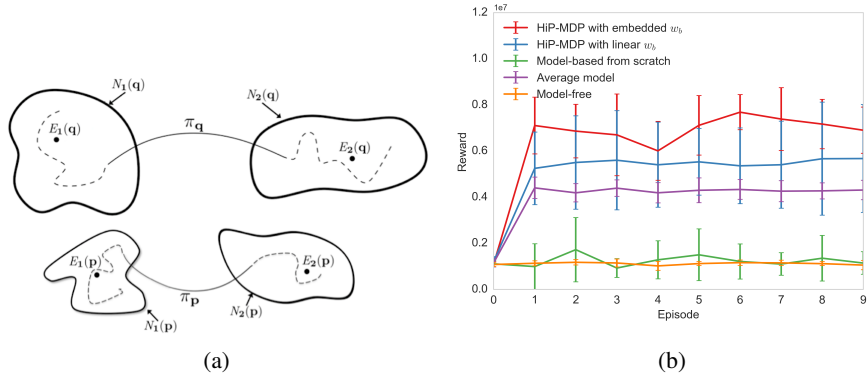

(a)                                                  (b)

Figure 4: (a) a visual representation of a patient with HIV transitioning from an unhealthy steady state to a healthy steady state using a proper treatment schedule, (b) a comparison of learning a policy for a new task instance $b$ using the HiP-MDP versus four benchmarks.

for more details.) The objective is to develop a treatment sequence that transitions the patient from an unhealthy steady state to a healthy steady state (Figure 4a, see Adams et al. [1] for a more thorough explanation). Small changes made to these parameters can greatly effect the behavior of the system and therefore introduce separate steady state regions that require unique policies to transition between them.

Figure 4b shows that the HiP-MDP develops an optimal control policy after a single episode, learning an unmatched optimal policy in the shortest time. The HIV simulator is the most complex of our three domains, and the separation between each benchmark is more pronounced. Modeling a HIV dynamical system from scratch from a single episode of observations proved to be infeasible. The average model, which has been trained off a large batch of observations from related dynamical systems, learns a better policy. The HiP-MDP with linear $w_b$ is able to transfer knowledge from previous task instances and quickly learn the latent parameterization for this new instance, leading to an even better policy. However, the dynamical system contains nonlinear interactions between the latent parameters and the state space. Unlike the HiP-MDP with embedded $w_b$, the HiP-MDP with linear $w_b$ is unable to model those interactions. This demonstrates the superiority of the HiP-MDP with embedded $w_b$ for efficiently transferring knowledge between instances in highly complex domains.

## 6 Related Work

There has been a large body of work on solving single POMDP models efficiently [6, 16, 24, 37, 45]. In contrast, transfer learning approaches leverage training done on one task to perform related tasks. Strategies for transfer learning include: latent variable models, reusing pre-trained model parameters, and learning a mapping between separate tasks (see review in [43]).

Our work falls into the latent variable model category. Using latent representation to relate tasks has been particularly popular in robotics where similar physical movements can be exploited across a variety of tasks and platforms [10, 20]. In Chen et al. [8], these latent representations are encoded as separate MDPs with an accompanying index that an agent learns while adapting to observed variations in the environment. Bai et al. [3] take a closely related approach to our updated formulation of the HiP-MDP by incorporating estimates of unknown or partially observed parameters of a known environmental model and refining those estimates using model-based Bayesian RL. The core difference between this and our work is that we learn the transition model and the observed variations directly from the data while Bai et al. [3] assume it is given and the specific variations of the parameters are learned. Also related are multi-task approaches that train a single model for multiple tasks simultaneously [5, 7]. Finally, there have been many applications of reinforcement learning (e.g. [32, 40, 44]) and transfer learning in the healthcare domain by identifying subgroups with similar response (e.g. [23, 28, 39]).

More broadly, BNNs are powerful probabilistic inference models that allow for the estimation of stochastic dynamical systems [11, 18]. Core to this functionality is their ability to represent both model uncertainty and transition stochasticity [25]. Recent work decomposes these two forms of uncertainty to isolate the separate streams of information to improve learning. Our use of fixed latent variables as input to a BNN helps account for model uncertainty when transferring the pretrained BNN to a new instance of a task. Other approaches use stochastic latent variable inputs to introduce transition stochasticity [12, 30].

We view the HiP-MDP with latent embedding as a methodology that can facilitate personalization and do so robustly as it transfers knowledge of prior observations to the current instance. This approach can be especially useful in extending personalized care to groups of patients with similar diagnoses, but can also be extended to any control system where variations may be present.

# 7 Discussion and Conclusion

We present a new formulation for transfer learning among related tasks with similar, but not identical dynamics, within the HiP-MDP framework. Our approach leverages a latent embedding—learned and optimized in an online fashion—to approximate the true dynamics of a task. Our adjustment to the HiP-MDP provides robust and efficient learning when faced with varied dynamical systems, unique from those previously learned. It is able, by virtue of transfer learning, to rapidly determine optimal control policies when faced with a unique instance.

The results in this work assume the presence of a large batch of already-collected data. This setting is common in many industrial and health domains, where there may be months, sometimes years, worth of operations data on plant function, product performance, or patient health. Even with large batches, each new instance still requires collapsing the uncertainty around the instance-specific parameters in order to quickly perform well on the task. In Section 5, we used a batch of transition data from multiple instances of a task—without any artificial exploration procedure—to train the BNN and learn the latent parameterizations. Seeded with data from diverse task instances, the BNN and latent parameters accounted for the variation between instances.

While we were primarily interested in settings where batches of observational data exist, one might also be interested in more traditional settings in which the first instance is completely new, the second instance only has information from the first, etc. In our initial explorations, we found that one can indeed learn the BNN in an online manner for simpler domains. However, even with simple domains, the model-selection problem becomes more challenging: an overly expressive BNN can overfit to the first few instances, and have a hard time adapting when it sees data from an instance with very different dynamics. Model-selection approaches to allow the BNN to learn online, starting from scratch, is an interesting future research direction.

Another interesting extension is rapidly identifying the latent $w_b$. Exploration to identify $w_b$ would supply the dynamical model with the data from the regions of domain with the largest uncertainty. This could lead to a more accurate latent representation of the observed dynamics while also improving the overall accuracy of the transition model. Also, we found training a DQN requires careful exploration strategies. When exploration is constrained too early, the DQN quickly converges to a suboptimal, deterministic policy—often choosing the same action at each step. Training a DQN along the BNN's trajectories of least certainty could lead to improved coverage of the domain and result in more robust policies. The development of effective policies would be greatly accelerated if exploration were more robust and stable. One could also use the hidden parameters $w_b$ to learn a policy directly.

Recognizing structure, through latent embeddings, between task variations enables a form of transfer learning that is both robust and efficient. Our extension of the HiP-MDP demonstrates how embedding a low-dimensional latent representation with the input of an approximate dynamical model facilitates transfer and results in a more accurate model of a complex dynamical system, as interactions between the input state and the latent representation are modeled naturally. We also model correlations in the output dimensions by replacing the GP basis functions of the original HiP-MDP formulation with a BNN. The BNN transition function scales *significantly* better to larger and more complex problems. Our improvements to the HiP-MDP provide a foundation for robust and efficient transfer learning. Future improvements to this work will contribute to a general transfer learning framework capable of addressing the most nuanced and complex control problems.

**Acknowledgements**    We thank Mike Hughes, Andrew Miller, Jessica Forde, and Andrew Ross for their helpful conversations. TWK was supported by the MIT Lincoln Laboratory Lincoln Scholars Program. GDK is supported in part by the NIH R01MH109177. The content of this work is solely the responsibility of the authors and does not necessarily represent the official views of the NIH.

## Footnotes

[3]We present a simplified version that omits their filtering variables $z_{kad} \in \{0, 1\}$ to make the parallels between our formulation and the original more explicit; our simplification does not change any key properties.

[4]Example code for training and evaluating a HiP-MDP, including the simulators used in this section, can be found at `http://github.com/dtak/hip-mdp-public`.

[5]The centers of mass and moments of inertia can also be varied. For our purposes we left them unperturbed.

[6]These markers are: the viral load ($V$), the number of healthy and infected CD4$^+$ T-lymphocytes ($T_1$, $T_1^*$, respectively), the number of healthy and infected macrophages ($T_2$, $T_2^*$, respectively), and the number of HIV-specific cytotoxic T-cells ($E$).

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
