[Supplementary Material · final_supplement.pdf]

# Appendix

## A  BNN-based transition functions with embedded latent weighting

### A.1  Scalability of BNN vs. GP based transition approximation

In this section, we demonstrate the computational motivation to replace the GP basis functions of the original HiP-MDP model [14] with a single, stand-alone BNN as discussed in Sec. 3) using the 2D navigation domain. To fully motivate the replacement, we altered the GP-based model to accept the latent parameters $w_b$ and a one-hot encoded action as additional inputs to the transition model. This was done to investigate how the performance of the GP would scale with a higher input dimension; the original formulation of the HiP-MDP [14] uses 2 input dimensions our proposed reformulation of the HiP-MDP uses 11 (2 from the state, 4 from the action and 5 from latent parameterization).

Figure 1: Here we show that the time to run an episode where the approximated transition model $\hat{T}$ and latent parameters $w_b$ are updated every 10 episodes. In the 2D navigation domain, the completion time is relatively constant for the BNN, whereas the GP's completion time drastically increases as more data is collected to construct the transition model.

We directly compare the run-time performance of training both the GP-based and BNN-based HiP-MDP over 6 unique instances of the toy domain, with 50 episodes per instance. Figure 1 shows the running times (in seconds) for each episode of the GP-based HiP-MDP and the BNN-based HiP-MDP, with the transition model $\hat{T}$ and latent parameters $w_b$ being updated after every 10 episodes. In stark contrast with the increase in computation for the GP-based HiP-MDP, the BNN-based HiP-MDP has no increase in computation time as more data and further training is encountered. Training the BNN over the course of 300 separate episodes in the 2D toy domain was completed in a little more than 8 hours. In contrast, the GP-based HiP-MDP, trained on the 2D toy domain, took close to 70 hours to complete training on the same number of episodes.

This significant increase in computation time using the GP-based HiP-MDP, on a relatively simple domain, prevented us from performing comparisons to the GP model on our other domains. (We do realize that there is a large literature on making GPs more computationally efficient [13, 34, 41]; we chose to use BNNs because if we were going to make many inference approximations, it seemed reasonable to turn to a model that can easily capture heteroskedastic, multi-modal noise and correlated outputs.)

### A.2  Prediction performance: benefit of embedding the latent parameters

In the previous section, we justified replacing the GP basis functions in the HiP-MDP in favor of a BNN. In this section, we investigate the prediction performance of various BNN models to determine

whether the latent embedding provides the desired effect of more robust and efficient transfer. The BNN models we will characterize here are those presented in Sec. 5 used as baseline comparisons to the HiP-MDP with embedded latent parameters. These models are:

- HiP-MDP with embedded $w_b$
- HiP-MDP with linear $w_b$
- BNN learned from scratch, without any latent characterization of the dynamics
- Average model: BNN trained on all data without any latent characterization of the dynamics.

For each of these benchmarks except the BNN trained from scratch, a batch of transition data from previously observed instances was used to pre-train the BNN (and learn the latent parameters for the HiP-MDPs). Each method was then used to learn the dynamics of a new, previously unobserved instance. After the first episode in the newly encountered instance, the BNN is updated. In the two models that use a latent estimation of the environment, the latent parameters are also updated. As can be seen in Fig 2, the models using a latent parameterization improve greatly after those first network and latent parameter updates. The other two (the model from scratch and average model) also improve, but only marginally. The average model is unable to account for the different dynamics of the new instance, and the model trained from scratch does not have enough observed transition data from the new instance to construct an accurate representation of the transition dynamics.

(a) 2D Toy Domain $\left[\mathcal{S} \in \mathbb{R}^2\right]$    (b) HIV Treatment domain $\left[\mathcal{S} \in \mathbb{R}^6\right]$

Figure 2: Comparison of HiP-MDP transition model accuracy with the transition models trained for the baselines presented in Sec. 5
.

The superior predictive performance of the two models that learn and utilize a latent estimate of the underlying dynamics of the environment reinforces the intent of the HiP-MDP as a latent variable model. That is, by estimating and employing a latent estimate of the environment, one may robustly transfer trained transition models to previously unseen instances. Further, as is shown across both domains represented in Fig. 2, the BNN with the latent parametrization embedded with the input is more reliably accurate over the duration the model interacts with a new environment. This is because the HiP-MDP with embedded latent parameters can model nonlinear interactions between the latent parameters and the state and the HiPMDP with linear latent parameters cannot. Moreover, the 2D navigation domain was constructed such that the true transition function is a nonlinear function of the latent parameter and the state. Therefore, the most accurate predictions can only be made with an approximate transition function that can model those nonlinear interactions. Hence, the 2D navigation domain demonstrates the importance of embedding the latent parameters $w_b$ with the input of the transition model.

## B    Experimental Domains

This section outlines the nonlinear dynamical systems that define the experimental domains investigated in Sec. 5. Here we outline the equations of motion, the hidden parameters dictating the

dynamics of that motion, and the procedures used to perturb those parameters to produce subtle variations in the environmental dynamics. Other domain specific settings such as the length of an episode are also presented.

## B.1   2D Navigation Domain

As was presented in Sec. 3 the transition dynamics of the 2D navigation domain follow:

$$\Delta x = (-1)^{\theta_b} c\big(a_x - (1 - \theta_b)\beta\sqrt{(x+1.5)^2 + (y+1.5)^2}\big)$$
$$\Delta y = (-1)^{\theta_b} c\big(a_y - \theta_b\beta\sqrt{(x+1.5)^2 + (y+1.5)^2}\big)$$
$$a_x = \begin{cases} 1 & a \in \{E, W\} \\ 0 & \text{otherwise} \end{cases}$$
$$a_y = \begin{cases} 1 & a \in \{N, S\} \\ 0 & \text{otherwise} \end{cases}$$

Where $c = 0.3$ and $\beta = 0.23$ are hyperparameters that restrict the agent's movement either laterally or vertically, depending on the hidden parameter $\theta_b$. In this domain, this hidden parameter is simply a binary choice ($\theta_b \in \{0, 1\}$) between the two classes of agent ("blue" or "red"). This force, used to counteract, or accentuate, certain actions of the agent is scaled nonlinearly by the distance the agent moves away from the center of the region of origin.

The agent accumulates a small negative reward (-0.1) for each step taken, a large penalty (-5) if the agent hits a wall or attempts to cross into the goal region over the wrong boundary. The agent receives a substantial reward (1000) once it successfully navigates to the goal region over the correct boundary. This value was purposefully set to be large so as to encourage the agent to more rapidly enter the goal region and move against the force pushing the agent away from the goal region.

At the initialization of a new episode, the class of the agent is chosen with uniform probability and the starting state of the agent is randomly chosen to lie in the region $[-1.75, -1.25]^2$.

## B.2   Acrobot

The acrobot domain [42] is dictated by the following dynamical system evolving the state parameters $s = \left[\theta_1, \theta_2, \dot\theta_1, \dot\theta_2\right]$:

$$\ddot\theta_1 = -d_1^{-1}(d_2\ddot\theta_2 + \phi_1)$$
$$\ddot\theta_2 = \left(m_2 l_{c2}^2 + I_2 - \frac{d_2^2}{d_1}\right)^{-1}\left(\tau + \frac{d_2}{d_1}\phi_1 - m_2 l_1 l_{c2}\dot\theta_1^2 \sin\theta_2 - \phi_2\right)$$
$$d_1 = m_1 l_{c1}^2 + m_2(l_1^2 + l_{c2}^2 + 2l_1 l_{c2}\cos\theta_2) + I_1 + I_2$$
$$d_2 = m_2(l_{c2}^2 + l_1 l_{c2}\cos\theta_2) + I_2$$
$$\phi_1 = -m_2 l_1 l_{c2}\dot\theta_2^2 \sin\theta_2 - 2m_2 l_1 l_{c2}\dot\theta_2\dot\theta_1 \sin\theta_2 + (m_1 l_{c1} + m_2 l_1)g\cos(\theta_1 - \pi/2) + phi_2$$
$$\phi_2 = m_2 l_{c2}g\cos(\theta_1 + \theta_2 - \pi/2).$$

With reward function $R(s, a) = -0.05\left([-l_1\cos(\theta_1) - l_2\cos(\theta_1 + \theta_2)] - l_1\right)^2$ if the foot of the pendulum has not exceeded the goal height. If it has, then $R(s, a) = 10$ and the episode ends. The hyperparameter settings are $l_{c1} = l_{c2} = 0.5$ (lengths to center of mass of links), $I_1 = I_2 = 1$ (moments of inertia of links) and $g = 9.8$ (gravity). The hidden parameters $\theta_b$ are the lengths and masses of the two links $(l_1, l_2, m_1, m_2)$ all set to 1 initially. In order to observe varied dynamics from this system we perturb $\theta_b$ by adding Gaussian ($\mathcal{N}(0, 0.25)$) noise to each parameter independently at the initialization of a new instance. The possible state values for the angular velocities of the pendulum are constrained to $\dot\theta_1 \in [-4\pi, 4\pi]$ and $\dot\theta_2 \in [-9\pi, 9\pi]$.

At the initialization of a new episode the agent's state is initialized to $s = (0, 0, 0, 0)$ and perturbed by some small uniformly distributed noise in each dimension. The agent is then free to apply torques to the hinge, until it raises the foot of the pendulum above the goal height or after 400 time steps.

## B.3   HIV Treatment

The dynamical system used to simulate a patient's response to HIV treatments was formulated in [1]. The equations are highly nonlinear in the parameters and are used to track the evolution of six core markers used to infer a patient's overall health. These markers are, the viral load ($V$), the number of healthy and infected CD4$^+$ T-lymphocytes ($T_1$ and $T_1^*$, respectively), the number of healthy and infected macrophages ($T_2$ and $T_2^*$, respectively), and the number of HIV-specific cytotoxic T-cells ($E$). Thus, $s = (V, T_1, T_2, T_1^*, T_2^*, E)$. The system of equations is defined as:

$$\dot{T}_1 = \lambda_1 - d_1 T_1 - (1 - \epsilon_1)k_1 V T_1$$
$$\dot{T}_2 = \lambda_2 - d_2 T_2 - (1 - f\epsilon_1)k_2 V T_2$$
$$\dot{T}_1^* = (1 - \epsilon_1)k_1 V T_1 - \delta T_1^* - m_1 E T_2^*$$
$$\dot{T}_2^* = (1 - \epsilon_2)N_T \delta(T_1^* + T_2^*) - cV - [(1 - \epsilon_1)\rho_1 k_1 T_1 + (1 - f\epsilon_1)\rho_2 k_2 T_2]V$$
$$\dot{E} = \lambda_E + \frac{b_E(T_1^* + T_2^*)}{(T_1^* + T_2^*) + K_b}E - \frac{d_E(T_1^* + T_2^*)}{(T_1^* + T_2^*) + K_d}E - \delta_E E.$$

With reward function $R(s, a) = -0.1V - 2e^4\epsilon_1^2 - 2e^3\epsilon_2^2 + 1e^3 E$, where $\epsilon_1, \epsilon_2$ are treatment specific parameters, selected by the prescribed action.

The hidden parameters $\theta_b$ with their baseline settings [1] are shown in Fig. 3.

| parameter | value | units | description |
|---|---|---|---|
| $\lambda_1$ | 10,000 | $\frac{cells}{mL \cdot day}$ | target cell type 1 production (source) rate |
| $d_1$ | 0.01** | $\frac{1}{day}$ | target cell type 1 death rate |
| $\epsilon_1$ | $\in [0, 1)$ | – | efficacy of reverse transcriptase inhibitor |
| $\epsilon_2$ | $\in [0, 1)$ | – | efficacy of protease inhibitor |
| $k_1$ | $8.0 \times 10^{-7}$ | $\frac{mL}{virions \cdot day}$ | population 1 infection rate |
| $\lambda_2$ | 31.98 | $\frac{cells}{mL \cdot day}$ | target cell type 2 production (source) rate |
| $d_2$ | 0.01** | $\frac{1}{day}$ | target cell type 2 death rate |
| $f$ | 0.34 ($\in [0, 1]$) | – | treatment efficacy reduction in population 2 |
| $k_2$ | $1 \times 10^{-4}$ | $\frac{mL}{virions \cdot day}$ | population 2 infection rate |
| $\delta$ | 0.7* | $\frac{1}{day}$ | infected cell death rate |
| $m_1$ | $1.0 \times 10^{-5}$ | $\frac{mL}{cells \cdot day}$ | immune-induced clearance rate for population 1 |
| $m_2$ | $1.0 \times 10^{-5}$ | $\frac{mL}{cells \cdot day}$ | immune-induced clearance rate for population 2 |
| $N_T$ | 100* | $\frac{virions}{cell}$ | virions produced per infected cell |
| $c$ | 13* | $\frac{1}{day}$ | virus natural death rate |
| $\rho_1$ | 1 | $\frac{virions}{cell}$ | average number virions infecting a type 1 cell |
| $\rho_2$ | 1 | $\frac{virions}{cell}$ | average number virions infecting a type 2 cell |
| $\lambda_E$ | 1 | $\frac{cells}{mL \cdot day}$ | immune effector production (source) rate |
| $b_E$ | 0.3 | $\frac{1}{day}$ | maximum birth rate for immune effectors |
| $K_b$ | 100 | $\frac{cells}{mL}$ | saturation constant for immune effector birth |
| $d_E$ | 0.25 | $\frac{1}{day}$ | maximum death rate for immune effectors |
| $K_d$ | 500 | $\frac{cells}{mL}$ | saturation constant for immune effector death |
| $\delta_E$ | 0.1* | $\frac{1}{day}$ | natural death rate for immune effectors |

Table 1: Parameters used in model (2.1). Those in the top section of the table are taken directly from Callaway and Perelson. Parameters in the bottom section of the table are adapted from those in Bonhoeffer, *et al.*. The superscripts * denote parameters the authors indicated were estimated from human data and ** denote those estimated from macaque data.

Figure 3: The hidden parameters that dictate the system dynamics of the HIV Treatment domain with their baseline values. Table courtesy of Adams et al. [1].

As was done with the Acrobot at the initialization of a new instance, these hidden parameters are perturbed with by some Gaussian noise ($\mathcal{N}(\theta_{b,i}, 0.25)$) each parameter independently. These perturbations were applied naively and at times would cause the dynamical system to lose stability or otherwise provide non-physical behavior. We filter out such instantiations of the domain and deploy the HiP-MDP on well-behaved and controllable versions of this dynamical system.

At the initialization of a new episode the agent is started at an unhealthy steady state

$$s = [163573, 5, 11945, 46, 63919, 24],$$

where the viral load and number of infected cells are much higher than the number of virus fighting T-cells. An episode is characterized by 200 time steps where, dynamically, one time step is equivalent to 5 days. At each 5 day interval, the patient's state is taken and is prescribed a treatment until the treatment period (or 1000 days) has been completed.

## C   Experiment Specifications

### C.1   Bayesian neural network

**HiP-MDP architecture**   For all domains, we model the dynamics using a feed-forward BNN. For the toy example, we used 3 fully connected hidden layers with 25 hidden units each, and for the acrobot and HIV domains, we used 2 fully connected hidden layers with 32 units each. We used rectifier activation functions, $\phi(x) = \max(x, 0)$, on each hidden layer, and the identity activation function, $\phi(x) = x$, on the output layer. For the HiP-MDP with embedded $w_b$, the input to the BNN is a vector of length $D + |A| + |w_b|$ consisting of the state $s$, a one-hot encoding of the action $a$, and the latent embedding $w_b$. The BNN architecture for the the HiP-MDP with linear $w_b$ uses a different input layer and output layer. The BNN input does not include the latent parameters. Rather the BNN output, $\hat{T}^{(BNN)}(s, a)$ is a matrix of shape $|w_b| \times D$. The next state is computed as $s' = w_b^T \hat{T}^{(BNN)}(s, a)$. In all experiments, the BNN output is the state difference $(s' - s)$ rather than the next state $s'$.

**Hyperparameters and Training**   For all domains, we put zero mean priors on the random input noise and the network weights with variances of 1.0 and $e^{-10}$, respectively, following the procedure used by Hernández-Lobato et al. [22]. In our experiments, we found the BNN performed best when initialized with a small prior variance on the network weights that increases over training, rather than using a large prior variance. Following Hernández-Lobato et al. [22], we learn the network parameters by minimizing the $\alpha$-divergence using ADAM with $\alpha = 0.5$ for acrobot and the toy example and $\alpha = 0.45$ for the HIV domain. In each update to the BNN, we performed 100 epochs of ADAM, where in each epoch we sampled 160 transitions from a prioritized experience buffer and divided those transitions into mini batches of size 32. We used a learning rate of 2.5E-4 for HIV and acrobot and learning rate of 5E-5 for the toy example.

The BNN and latent parameters were learned from a batch of transition data gathered from multiple instances across 500 episodes per instance. For the toy example, acrobot, and HIV, we use data from 2, 8, and 5 instances, respectively. For HIV, we found performance improved by standardizing the observed states to have zero mean, unit variance.

### C.2   Latent Parameters

For all domains, we used $|w_b| = 5$ latent parameters. The latent parameters were updated using the same update procedure as for updating the BNN network parameters (except with the BNN network parameters held fixed) with a learning rate of 5E-4.

### C.3   Deep Q-Network

To learn a policy for a new task instance, we use a Double Deep Q Network with two full connected hidden layers with 256 and 512 hidden units, respectively. Rectifier activation functions are used for the hidden layers and the identity function is used on for the output layer. For all domains, we update the primary network weights every $N_\pi = 10$ time steps using ADAM with a learning rate of 5E-4, and slowly update the target network to mirror the primary network with a rate of $\tau = 0.005$. Additionally, we clip gradients such that the L2-norm is less than 2.5. We use an $\epsilon$-greedy policy starting with $\epsilon = 1.0$ and decaying $\epsilon$ after each episode (each real episode for the model-free approach and each approximated episode for the model-based approaches) with a rate of 0.995.

In model-based approaches, we found that the DQN learns more robust policies (both on the BNN approximate dynamics and the real environment) from training exclusively off the approximated transitions of the BNN. After training the BNN off the first episode, we train the DQN using an initial batch of $N_f = 500$ approximated episodes generated using the BNN.

## C.4 Prioritized Experience Replay Buffers

We used a TD-error-based prioritized experience replay buffer [38] to store experiences used to train the DQN. For model-based approaches, we used a separate squared-error-based prioritized buffer to store experiences used to train the BNN and learn the latent parameterization. Each prioritized buffer was large enough to store all experiences. We used a prioritization exponent of 0.2 and an importance sampling exponent of 0.1.

## D  Long run demonstration of policy learning

We demonstrate that all benchmark methods used learn good control policies for new, unique instances of the acrobot domain (Figure 4a) and the HIV treatment domain (Figure 4b) with a sufficient number of training episodes. However, in terms of policy learning efficiency and proficiency, comparing the performance of the HiP-MDP with the benchmark over the first 10 episodes is instructive—as presented the Experiments section of the main paper. This format emphasizes the immediate returns of using the embedded latent parameters to transfer previously learned information when encountering a new instance of a task.

(a) Acrobot　　　　　　　　　　　　　　　(b) HIV Treatment

Figure 4: A comparison of learning a policy for a new task instance $b$ using the HiP-MDP versus four benchmarks over more episodes. The mean reward for each episode over 5 runs is shown for each benchmark. The error bars are omitted to show the results clearly.