[Reviews · NeurIPS 2017]

Reviewer 1



Summary: This paper presents a new transfer learning approach using Bayesian Neural Network in MDPs. They are building on the existing framework of Hidden Parameter MDPs, and replace the Gaussian process with BNNs, thereby also modeling the joint uncertainty in the latent weights and the state space. Overall, this proposed approach is sound, well developed and seems to help scale the inference. The authors have also shown that it works well by applying it to multiple domains. The paper is extremely well written. This paper would make an important addition to the literature of transfer learning within the deep reinforcement learning. Things I liked about the paper: - The paper is extremely well written and easy to understand even for someone not completely familiar with the area. - The use of BNNs is well justified and really seem to be helping to expand the scope of applications to more complex applications. - The experiments use multiple strong baselines and show clear improvements over them in multiple domains, which include both toy and real world domains. Concerns: - There isn't enough analysis of the space requirements for the proposed method, including for the global reply buffer. - The inference procedure proposed in this paper seems to have too many moving parts and possibly things to tune. It's not clear how much effort would be required to apply this method in a new domain - The experiments description is a bit short for each of the domain due to space constraints. Making it slightly harder to understand them. For example, it was not clear what the Wb's and rewards corresponded to in each of the application domains. Minor Edits: - Appendix references are missing in multiple places: line 114,198, and 215 - A stylistic suggestion: In the beginning of section 3, use present tense consistently instead of switching between past and present tense. - 79: allows us to model - 88: wall that is blocking - 97: model these both classes? - 99: depends on interactions - 100: a linear function - 122: stored in both - 146: developing a control policy

Reviewer 2



SUMMARY The authors consider the problem of learning multiple related tasks. It is assumed each tasks can be described by a low-dimensional vector. A multi-task neural network model can than be learned by treating this low-dimensional vector as additional input (in addition to state and action). During optimization, both these low-dimensional inputs and the network weights are optimized. The network weights are shared over all tasks. This procedure allows a new task to be modeled very quickly. Subsequently, 'virtual samples' taken from the model can be used to train a policy (using DDQN). The method is evaluated on three different task families. REACTION TO AUTHOR'S REPLY Thanks for the response. It cleared up many questions I had. I changed my rating accordingly. I think it would be good to show the standard deviation over independently trained models to show the variance induced (or absence of such variance) by the (random) choice of initial tasks and possibly the model training procedure. In the toy task, it seems there is always one 'red' and one 'blue' tasks in the initial data. Does that mean the initial tasks are hand-picked (rather than randomly drawn?). TECHNICAL QUALITY The used approach seems sensible for the task at hand. The development of the method section seems mostly unproblematic. Two issues / suggestions are 1) when tuning the model, shouldn't the update for the network weight take samples from the global buffer into account to avoid catastrophic forgetting for the next task? 2) I'm not sure why you need prioritization for the model update, if you are looking to minimize the mean squared error wouldn't you want to see all samples equally often? Support for the method is given by experimental evaluation. The method is mostly compared to ablated versions, and not to alternatives from the literature (although one of the methods is supposed to be analogous to the method in [11], and there might not be many other appropriate methods to compare to - would a comparison to [2] be possible?). In any case, the selection of methods to compare to includes different aspects, such as model-free vs model based, and independent models vs. multitask vs. single model for all tasks. It is not clear whether multiple independent runs were performed for each methods. Please specify this, and specify what the error bars represent (standard deviation or standard error, one or two deviations, spread between roll-outs or between independent trials, how many trials). Dependent on the experiment, the proposed method seems at least as good as most baselines, and (at least numerically) better than all baselines on the most challenging task. The additional material in the appendix also shows the runtime to be much faster compared to the GP-based method in [11]. NOVELTY The general framework is relatively close to [11]. However, the use of neural networks and the possibility to model non-linear dependence and covariance between states and hidden parameters seems like an important and interesting contribution. SIGNIFICANCE AND RELEVANCE Multi-task and transfer learning seem important components for RL approaches in application domains where rapid adaptation or sample efficiency are important. The proposed method is limited to domains where the difference between tasks can be captured in a low dimensional vector. CLARITY There are a couple of issues with the paper's clarity. If found Figure (1) hard to understand, given its early position in the text and the absense of axis labels. I also found Algorithm 1 hard to understand. The 'real' roll-out only seemed to be used in the first episode, what is done with the data after the first episode (apart from storing in global memory for next task?). Maybe split move entire if (i=0) out of the for loop (and remove the condition), than the forloop can just contain (15) if I'm not mistaken. The description of the experiments is missing too many details. I couldn't find in either appendix or paper how many source instances the method was trained on before evaulation on the target task for the second or third experiment. I also couldn't find how many times you independently trained the model or what error bars represent. Two source tasks for the toy tasks seems very little given that you have differences between instances across multiple axis - with just two examples would it only learn 'red' vs 'blue' since this is the most important difference? Furthermore there are quite a few spelling errors and similar issues in the text. I'll mention some of them below under minor comments. MINOR COMMENTS - Typesetting: subscripts GP and BNN are typeset as products of single-letter variables - You might want to explain the subscripts k,b,a,d in (1) briefly. - 79: model -> to model - since your model's epsilon distribution is not dependent on the current state, I don't think this could model hereskedasticyity (line 31). If so, please explain. (Uncertainty in weights lead to different predictive uncertainty in parts of the state-space, like a regurlar GP. But, like a regular GP this doesn't necessarily mean you model different 'noise distribution' or risk in different part of your state space ) - 'bounded number of parameters' seems somewhat ambiguous (do you mean a finite-dimensional vector, or finitely many values for this parameter?). - 81: GP's don't really care about high-d input representation (linear scaling in input dimensionality), tractability issues mainly concern number of samples (cubic scaling for naive implementations). - 92-93: are delta_x and delta_y really the same weather a=E or a=W as it appears from this equation? - 100: a linear of function -> a linear function - 114: Sec ?? - 123: is develop -> is to develop - 169 from which learn a policy -> please rephrase - 174 on trained two previous -> please rephrase - 188 "of average model"-> of the average model - 198 Appendix ?? - 230 - I'm not sure if embedded is the right word here - 247-249 - it would be interesting to see a comparison - 255 - "our work" - references: It would be good to go over all of these ones as there are many small issues. E.g. the IEEE conferences mention the year and 'IEEE' twice. Names like 'Bayesian' or 'Markov' and abbreviations like STI, HIV, POMDP, CSPBVI should be capitalized. Q-learning too. For [21] I would cite the published (ICLR) version. Publication venue is missing for [35]. - appendices: just wanted to make sure you really meant exp(-10), not 1E-10. Does the prior need to be so small? Why? What do you mean when you say the prior variance changes over training? I think with |w| you mean the number of elements, but you used notation of a norm (w is a vector, not a set).

Reviewer 3



This paper proposes a new formulation of Hidden Parameter Markov Decision Pro- cess (HiP-MDP) which is a model with a structure to adopt subtle change in tasks by using a latent task parametrization that is combined with state and action to define system dynamics. Differently from the previous HiP-MDP, the proposed method models the interaction between states and the latent parameters for HiP-MDP by using a Bayesian neural network that also improves scaling. The paper explains the motivation and contributions clearly. Integrating the interaction between latent parameters and states in HiP-MDP is an interesting motivation but authors need to demonstrate better that this extension yields a significant improvement in other different settings than toy data and HIV data. Rather than running linear version, direct comparison to Doshi-Velez and Konidaris would be better to show that the extension is significantly better by integrating the interaction properly . Authors state that closer work is Bai et al. however they do not compare to them. There are some typos to be fixed: The paragraph related to health care is incomplete:“diagnoses for each subgroup. Our work” and “Appendix ??” The paper proposes a novel approach for transfer learning among tasks with subtle changes by adopting HiP-MDP. In order to be convinced that the extension is significant, having more results demonstrating the accurate modeling of the interaction in HiP-MDP would be helpful.